# A Low-Cost Optomechatronic Diffuse Optical Mammography System for 3D Image Reconstruction: Proof of Concept

**DOI:** 10.3390/diagnostics15050584

**Published:** 2025-02-27

**Authors:** Josué D. Rivera-Fernández, Alfredo Hernández-Mendoza, Diego A. Fabila-Bustos, José M. de la Rosa-Vázquez, Macaria Hernández-Chávez, Gabriela de la Rosa-Gutierrez, Karen Roa-Tort

**Affiliations:** 1Laboratorio de Optomecatrónica y Energías, UPIIH, Instituto Politécnico Nacional, Distrito de Educación, Salud, Ciencia, Tecnología e Innovación, San Agustín Tlaxiaca 42162, Mexico; ahernandezm1926@alumno.ipn.mx (A.H.-M.); dfabilab@ipn.mx (D.A.F.-B.); mhernandezch@ipn.mx (M.H.-C.); 2Laboratorio de Biofotónica, ESIME-Zac, Instituto Politécnico Nacional, Gustavo A. Madero, Mexico City 07320, Mexico; mdelaros@ipn.mx; 3Hospital de Especialidades del Niño y la Mujer, Av. Luis M. Vega y Monroy, Querétaro 1000, Mexico; gabymed_hjm@hotmail.com

**Keywords:** optical imaging, breast cancer diagnosis, 3D reconstruction, optomechatronic system, diffuse optical mammography

## Abstract

**Background**: The development and initial testing of an optomechatronic system for the reconstruction of three-dimensional (3D) images to identify abnormalities in breast tissue and assist in the diagnosis of breast cancer is presented. **Methods**: This system combines 3D reconstruction technology with diffuse optical mammography (DOM) to offer a detecting tool that complements and assists medical diagnosis. DOM analyzes tissue properties with light, detecting density and composition variations. Integrating 3D reconstruction enables detailed visualization for precise tumor localization and sizing, offering more information than traditional methods. This technological combination enables more accurate, earlier diagnoses and helps plan effective treatments by understanding the patient’s anatomy and tumor location. **Results**: Using Chinese ink, it was possible to identify simulated abnormalities of 10, 15, and 20 mm in diameter in breast tissue phantoms from cosmetic surgery. **Conclusions**: Data can be processed using algorithms to generate three-dimensional images, providing a non-invasive and safe approach for detecting anomalies. Currently, the system is in a pilot testing phase using breast tissue phantoms, enabling the evaluation of its accuracy and functionality before application in clinical studies.

## 1. Introduction

In medicine, the early and precise diagnosis of diverse conditions is paramount [1]. The expeditiousness of diagnosis holds the potential to markedly influence mortality rates, even in diseases such as breast cancer [2]. Breast cancer is the most frequent neoplasm in women worldwide [3] and, according to Breast Cancer Statistics and Resources, it is estimated that every 14 s, a new case is diagnosticated somewhere across the globe [4]. In 2022, a total of 2.3 million women were diagnosed with breast cancer and there were 670,000 deaths related to breast cancer globally; while it is widely acknowledged that being female constitutes the most significant risk factor for breast cancer, it is noteworthy that approximately 0.5–1% of breast cancer cases are diagnosed in men [5]. This condition can be effectively treated in approximately 70–80% of patients diagnosed with early-stage non-metastatic disease [3].

Currently, in developing countries such as Mexico, the diagnosis of breast cancer is based on conventional methodologies, including patient self-examination or medical palpation, ultrasound, mammography, magnetic resonance imaging, and even biopsy [6]. Among these methodologies, X-ray mammography is regarded as the gold-standard diagnostic tool for early-stage detection. Mammography is a widely recognized biomedical imaging technique employed in the screening of breast cancer [7].

Several studies aim to integrate biomedical images from mammograms, as well as ultrasound images in many cases, with machine learning and deep learning systems to identify abnormalities in tissue and to provide diagnoses based on the analysis of these images [7,8,9,10,11]. However, the use of conventional mammography involves the application of ionizing radiation, which, in high doses or prolonged exposure, can cause severe damage and even death [12,13]. Furthermore, the maintenance of mammography systems in developing countries poses significant challenges, resulting in substantial costs and, consequently, an increase in the price of the examination.

On the other hand, diffuse reflectance spectroscopy [14], diffuse optical mammography or tomography, and other optical techniques using new instrumentation represent highly promising non-invasive techniques for the medical diagnosis of breast cancer [15,16].

Biomedical optical techniques enable non-invasive tissue diagnosis and monitoring through light–tissue interactions. Optical spectroscopy analyzes absorption and scattering to detect biomarkers without the need for biopsies. Diffuse optical tomography (DOT) and optical coherence tomography (OCT) provide depth-resolved and high-resolution imaging, respectively, without ionizing radiation. Raman spectroscopy and fluorescence imaging offer high molecular specificity, while photoacoustic imaging (PAT) combines light and ultrasound to visualize vascular structures and to assess tissue oxygenation [14,15,16,17,18].

These techniques enhance conventional methods by enabling early disease detection and non-invasive biochemical analysis. Their integration into clinical practice strengthens early detection, reduces the need for invasive procedures, and improves diagnostic accuracy. Table 1 presents various optical techniques for medical diagnosis, their principles of operation, and their advantages over traditional techniques.

In this manuscript, the development and first tests of an optomechatronic system for diffuse optical mammography, combined with 3D reconstruction techniques, for the identification of abnormalities in breast tissue is presented. This technique is proposed as a complementary auxiliary tool for medical specialists.

## 2. Materials and Methods

Optomechatronic systems in the diagnosis of breast cancer can provide non-invasive, precise, and efficient evaluation methods [17]. These systems integrate optics and mechatronics to generate detailed images of breast tissue, facilitating the early detection of abnormalities [18]. Incorporating three-dimensional reconstruction techniques further enhances the capability of identifying and characterizing potential tumors, offering medical professionals an advanced tool for diagnosing and monitoring breast cancer abnormalities [19]. Additionally, as a non-invasive technique, optomechatronic systems reduce risks and discomfort to patients, which can lead to increased acceptance and frequency of screening exams [20].

### 2.1. Instrumentation

Here, the development of a 3D diffuse optical mammography reconstruction optomechatronic system is presented. The system involves the integration of mechanical and electronic control systems and the inclusion of optical components.

In Figure 1, a general view of the system is depicted. The device consists of an Intel© D457 camera with RealSense™ technology for acquiring images of the tissue. It includes a mechanism that allows the camera to rotate to obtain multiple captures. Additionally, the equipment features LED lighting, arranged to provide uniform illumination over the sample of interest. This entire system is controlled by a Jetson Nano™ embedded board, enabling wireless communication through a graphical user interface (GUI).

As a result, the 3D DOM reconstruction system can be described in terms of three main areas: the optics applied in the DOM technique, the physical system comprising both the mechanical and electronic control components, and the three-dimensional reconstruction phase for obtaining a breast tissue model from a phantom.

### 2.2. Diffuse Optical Mammography (DOM)

DOM is an optical technique that involves the implementation of a continuous or pulsed light source that irradiates the breast tissue to enable light measurements for obtaining the absorption and scattering tissue coefficients [21,22] or for generating digital images for computer-vision pattern recognition [23].

The interaction between the light and the tissue occurs through absorption, scattering, and reflection processes. Chromophores, which are molecules or functional groups capable of absorbing light at specific wavelengths, such as hemoglobin, water, and melanin, absorb light and convert it into heat. Scattering occurs when subcellular structures, such as collagen fibers and cells, deflect photons, altering how light is distributed within the tissue. Reflection takes place when light is reflected at the interface between two media with different refractive indices, such as between skin and air. Light penetration varies depending on the wavelength, composition, and density of the tissue, with longer wavelengths penetrating more deeply [24]. These phenomena are fundamental to medical technologies such as diffuse optical mammography (DOM) and optical spectroscopy, which exploit the optical properties of tissues to generate non-invasive images.

In particular, in the study of breast cancer, it is known that a tissue with lesions (presence of abnormalities or tumors) shows a higher blood volume, i.e., higher hemoglobin (Hb) concentration, implying an increased absorption of red or infrared light, increased water content (H_2_O) in the tissue, and decreased lipid concentration compared with healthy breast tissue [25].

Generally, when implementing optical techniques, the aim is to obtain the optical parameters of the study sample. In the case of diffuse optical imaging systems, the light absorption distribution is typically obtained by employing multiple light sources and detectors around the breast [26] or by using transmission light techniques with a light source and detector on opposite sides of the tissue to measure light absorption through the tissue to identify the presence of any abnormalities [27].

In this research, the developed system aims to provide visual information to the specialist for medical diagnosis through three-dimensional tissue reconstruction. To achieve this, a transillumination configuration is implemented, which complements the DOM technique, focusing on obtaining reconstruction images rather than measuring the optical parameters of the tissue.

The implemented lighting system consists of a set of four QINGYING brand LEDs, each with 3 W of electrical power, a wavelength of 635 nm, and a viewing angle of 120°.

After characterizing the LEDs, it was decided to use four illuminants placed 25 cm above the phantom at an inclination of 60°. This arrangement was made to achieve uniform lighting of the sample under study. The configuration of the LEDs and the distribution of light in the sample, seen from a top view, is shown in Figure 2.

To carry out the irradiation process and to ensure that it is homogeneous over the phantom, in addition to characterizing the LEDs, it was necessary to construct a matte black cabin-like structure. This prevents reflections and isolates the measurement environment from the surroundings, ensuring that only the LED light interacts with the sample. In Figure 2, the black outline represents the walls of this cabin.

Having described the optical part of the system, it is crucial to understand how it interacts with the mechanical and electrical components to achieve seamless operation for the DOM implementation.

### 2.3. Mechanical and Electrical System

Mechanical and electronic components are elements that enable synergistic integration to create automated devices capable of performing tasks with high precision through some form of control [28]. The mechanical part includes all the components involved in the generation and transmission of movement and force. These components may include motors, gears, actuators, and mechanical structures. On the other hand, the electronic part includes all the components that supply and manage the electrical energy necessary for the system’s operation. These range from power supplies and batteries to control circuits and sensors [29]. The combination of these two areas, along with control and programming, allows the creation of advanced mechatronic systems that are capable of performing tasks autonomously and efficiently [30].

#### 2.3.1. Mechanical Components

For the arrangement of the physical system, it was considered that the implemented camera needs a minimum distance of 50 cm from the point of interest for a suitable focus, a controlled lighting environment, and the ability to place phantoms with different characteristics and dimensions, with a maximum diameter of 30 cm. Therefore, a booth with a base of 35 cm × 35 cm and a height of 70 cm was built, thus obtaining the booth structure shown in Figure 3A.

The structure contains a set of supports that allow the assembled LEDs to be mounted to their respective heatsinks. These pieces are positioned at each corner of the base of the structure, as shown in Figure 3B. Finally, at the top of the booth, a disk-type support base is placed, serving as the base for the electronic components and the image acquisition system. This is mounted, as shown in Figure 3C, onto a stepper motor that enables the system to capture photographs from different positions. It is worth noting that the motor allows rotation of the system every 30°.

The disk-type support base is made of aluminum 1060, with a diameter of 20 cm, capable of supporting both the image capture module and the electronics module. These modules include the Jetson Nano™ board, a printed circuit board (PCB) for electronic connections, antennas for wireless communication, and the D547 camera. A better view of this component is shown in Figure 4.

The above-described mechanical structure, depicted in Figure 4, is designed to support and protect the device’s electronics. Furthermore, the arrangement of each coupling and the general anchoring of all the device components facilitates heat dissipation, thereby enhancing its thermal efficiency.

#### 2.3.2. Electrical Components

In general, the device is powered by a switching power supply with three voltage outputs: 5 V @ 8 A, 12 V @ 8 A, and 24 V @ 3 A. The first output is used to power the digital electronics, the second for the LEDs and their drivers, and the last for the motor and its controller.

The central electronic component of the system, the Jetson Nano™ embedded board, allows for the control of the entire device and wireless communication with a graphical user interface. Regarding device control, the embedded system enables the manipulation of pulse-width modulation (PWM) signals to modulate the radiant flux of the LEDs.

The PWM signals generated from the embedded system are sent to the PCB of the electronic module, represented by the letter B in Figure 4, to activate the corresponding current drivers that power the LEDs. It is important to mention that this modulation is crucial, as changing the intensity of the lighting system allows for studies on different types of phantoms. Additionally, in the future implementation of this technique on patients, it will be possible to consider that the characteristics of each person’s tissue are different.

In addition to controlling the lighting, the central board activates the power stage that enables the operation of the stepper motor that rotates the capture module.

Finally, the Jetson Nano™ system enables the integration of the physical device with a user interface that facilitates control of the aforementioned parameters, including the rotation angle of the capture module and the radiant flux through pulse-width modulation (PWM) signals. This interface also allows for manipulation of the camera parameters, such as exposure and shutter speed, as well as the storage and visualization of captures throughout the study process. Additionally, the interface provides a three-dimensional reconstruction view of the phantom and manages all aspects related to its storage.

Figure 5 shows a diagrammatic representation of the electrical connections between the described elements.

### 2.4. 3D Reconstruction

In the technological context, three-dimensional reconstruction is understood as the systematic process of generating a digital representation of an object or scene from two-dimensional data such as images or photographs [31,32]. This process entails the application of techniques such as photogrammetry, instant neural graphics primitives, structure for motion, cloud points, and laser scanning, among others [33,34,35]. The objective is to create digital models that faithfully replicate the geometry, texture, and, in some cases, the color of the object or scene of interest. This aims to facilitate detailed analysis through an as-accurate-as-possible reproduction of the object.

The developed system uses the RealSense™ D457 camera, of Intel Corporation in the Santa Clara, CA, USA in combination with the Jetson Nano™ board to perform the 3D reconstruction. The process begins when the camera projects an infrared light pattern onto the scene. When it encounters surfaces at different distances, this pattern deforms, and the camera’s sensors capture these deformations to calculate depth using triangulation algorithms. Additionally, the camera captures high-resolution color images, which are integrated with the depth data to obtain a detailed and textured representation of the environment.

The camera data are sent to the Jetson Nano™ for processing, which receives the RGB images and depth maps, performs preprocessing to remove noise and to calibrate the data, and subsequently merges this information to create colored point clouds or textured 3D models.

To facilitate the interaction and visualization of these data, a graphical user interface (GUI) has been developed using the high-level Python language, which allows users to explore and analyze the results intuitively. This provides a range of tools and options for visualizing 3D models, as well as for manipulating the captured data. The GUI has five buttons, two sliders, a text entry field, and a camera display area, as shown in Figure 6.

The program’s algorithmic logic was designed so that the graphical interface allows for the following functions: starting, stopping, and adjusting the viewing angle; continuously displaying the camera feed; adjusting the light intensity and camera exposure; and, finally, saving the reconstructed model for subsequent visualization.

## 3. Results

The American Cancer Society classifies breast cancer stages using a T followed by a number from 0 to 4, which describes the main (primary) tumor size. Higher T numbers indicate a larger tumor and a wider spread to tissues near the breast. When a tumor is 20 mm (about 0.79 inches) or less in diameter, a doctor can classify it as an initial tumor (T1 level) [36]. Additionally, the diameters of the internal mammary vessels are approximately 2.5 mm (about 0.1 inches) [37].

A POLYTECH Health & Aesthetics GmbH Co (Dieburg, Germany) breast tissue prosthesis, with a thickness of 5 cm, commonly used in esthetic surgeries, was employed as a phantom. Inside this, three tumors of different diameters were simulated, all within the T1 range: 10 mm, 15 mm, and 20 mm. For this purpose, Chinese ink was used, which can be seen in Figure 7. Chinese ink was used because of its similar absorption coefficient in the 630 to 660 nm with tumors cancers [38].

The study of optical properties, such as the refractive index, is essential for understanding light–tissue interactions in biomedical applications. Real breast tissue exhibits a composite structure comprising water, fat, and other biological components, resulting in an effective refractive index that varies based on its composition. On the other hand, breast implants, which are primarily made of silicone or saline solution, have well-defined and relatively constant refractive indices. These properties play a crucial role in optical imaging, diagnostic techniques, and therapeutic applications. The following table, Table 2, provides a comparative overview of the refractive indices of real breast tissue and common breast implant materials, measured at a wavelength of 635 nm, which lies in the red region of the visible spectrum.

In this study, experiments were conducted on two types of phantoms: one designed to simulate vascularity, meaning veins and blood vessels, and another specifically developed for tumor simulation. This research presents the results obtained from the phantom utilized for tumor simulation.

For the data analysis and system validation, 50 images of the phantom were acquired, each comprising a total of 600 measurements. The irradiance was measured both on the surface of the sample and after transmission through it, enabling the calculation of the sample absorbance and transmittance. The results obtained were 0.6416 (64.16%) for transmittance and 0.1928 for absorbance. We considered that the intensity of the transmitted light after passing through the sample was 35.8 µW and that the intensity of the incident light on the sample was 55.8 µW. Based on these values, it was determined that the optical efficiency of the LED when interacting with the phantom was 17%. Values were computed using the PM125D radiometer from Thorlabs©. The transmittance and absorbance values were calculated using the mathematical models presented in Equations (1) and (2), respectively.(1)$$T=\frac{I}{I_{0}}$$
where T=Transmittance,
 I=Intensity of the transmitted light after passing through the sample, and ${\;I}_{o}=Intensity\;of\;the\;incident\;light\;on\;the\;sample$
(2)$$A=log\left(\frac{I}{I_{0}}\right)$$
where A=Absorbance, I=Intensity of the transmitted light after passing through the sample, and ${\;I}_{o}=Intensity\;of\;the\;incident\;light\;on\;the\;sample$


Based on the transmittance value, the absorption coefficient of the phantom used can be determined using the Beer–Lambert law, as outlined in the mathematical model of Equation (3).(3)$$T=e^{\mu_{a}d}$$
where T=Transmittance, ${\;\mu }_{a}=Absorption\;coefficient$, and d=Thickness of the material

From Equation (3), ${\;\mu }_{a}$ can be computed using Equation (4).(4)$$\mu_{a}=-\frac{ln(T)}{d}$$

In the present research, the calculated absorption coefficient was 0.0877 cm^−1^, which was similar to what Ismagilova et al. reported in [41] for silicone. A scattering coefficient of 41 cm⁻^1^ was reported, a value considered valid for this investigation due to the similarity of the obtained results.

It is important to mention that the reference indicates that the polymer is enriched with aluminum powder. In general, the addition of aluminum powder in silicone implants aims to improve the mechanical, thermal, biocompatible, and functional properties of the material, making it better suited to the specific needs of medicine and implant technology [42]. This suggests, based on the data found, that the implant used as a phantom most likely contains aluminum powder to enhance its biocompatibility properties, even though the manufacturer does not explicitly state this in the materials used.

Finally, the penetration depth achieved by the system was calculated using the mathematical model of Equation (5), resulting in a value of 11.27 cm.(5)$$\delta =\frac{1}{\mu_{a}}$$
where δ=Pentration depth

The penetration depth refers to the distance at which an electromagnetic wave (such as light, ultrasound, or a radio signal) can propagate within a material before its intensity is significantly reduced, typically to 37% (1/e) of its initial value due to absorption and scattering in the medium.

If a penetration depth of 11.27 cm is indicated, it means that after traveling this distance within the material, the wave’s intensity has decreased.

Following the measurements, the 50 acquired images were examined to identify the simulated lesions through pattern recognition techniques. This evaluation enabled the determination of the system’s specificity and sensitivity.

Table 3 presents the results obtained, along with a comparison with conventional techniques such as mammography and ultrasound. For this purpose, for each of the 50 images, one region without a lesion and one region with a lesion were selected to evaluate the false positive and false negative rates.

Based on the results presented in the table, the system identified 36 out of 50 cases with anomalies, while 14 were classified as false negatives. This limitation may be attributed to the anomaly being located at a greater depth, restricting light penetration, or to the presence of artifacts—distortions or undesired errors in the reconstructed image arising from limitations in the acquisition, processing, or data modeling process—which compromised the detection of small structures.

On the other hand, 34 out of 50 cases without anomalies were correctly identified, while 16 were false positives. This indicates that the system flagged areas as suspicious despite the absence of anomalies. A possible cause for this could be the optical properties of the material itself, such as absorption and scattering within the medium, or errors in the reconstruction algorithm. These issues could potentially be mitigated by using digital filters for preprocessing or postprocessing of the captured image.

The DOM with 3D image reconstruction technique performs reasonably well compared with mammography and ultrasound. Although it does not surpass ultrasound in terms of sensitivity, it offers a higher specificity than ultrasound and is comparable to mammography in minimizing false positives. Overall, DOM with 3D image reconstruction appears to be a promising technique, particularly given its ability to balance both sensitivity and specificity, though improvements in sensitivity could enhance its effectiveness.

Several tests were conducted by varying the camera exposure and the intensity of LED irradiation. Figure 8 shows some of these tests. It should be noted that the image shows a top view of the reconstruction to estimate the position of the abnormalities within the tissue.

It is important to highlight that, in addition to varying the exposure, tests were conducted by adjusting the illumination intensity. For the phantom used, the most favorable results were obtained with an intensity of 50%. Furthermore, it is noteworthy that the capture system was rotated to identify the position where the maximum amount of information was obtained.

It is observed in Figure 8 that, for a camera exposure of 1000, only two of the three anomalies, corresponding to those with diameters of 15 and 20 mm (i.e., the larger ones), can be clearly distinguished. In the second test, with an exposure of 2000, the three anomalies are identified. Finally, an exposure of 5000 saturated the images; however, this eliminates much of what would be considered healthy tissue and allows the identification of each anomaly within the tissue.

Additionally, the software of the system allows for the generation of three-dimensional models for more detailed analysis. An example of extracting 3D models from captures is illustrated in Figure 9.

An essential aspect of the obtained reconstruction is the consideration that the camera used has a spatial resolution of 1280 × 720 pixels. Based on this value and the number of pixels identified in the image, a relationship is established between the real size of the object and its pixel representation. In the images with an exposure of 1000, where tumors of 15 and 20 mm were identified, the measured sizes were 13 and 18 mm, respectively. In contrast, for the image with an exposure of 2000, the sizes of the anomalies were identified as 11, 16, and 21 mm for anomalies measuring 10, 15, and 20 mm, respectively. This indicates a good approximation of the actual size. However, regarding depth, further adjustments are required, as rotating the 3D image to establish a reference on the *Z*-axis introduces distortion due to the physical structure of the phantom. This distortion may cause additional variations in referencing, which necessitates attention in future experiments to enhance accuracy.

## 4. Discussion

Early detection of tumors in breast cancer is essential for minimizing mortality rates, as it facilitates precise and timely therapeutic interventions [44,45]. Furthermore, the application of optical imaging techniques represents a significant advancement in the early detection of breast cancer compared with traditional methods and biomarker-based techniques [46,47].

This study presents an optomechatronic system that integrates optical techniques, namely diffuse optical mammography (DOM) and three-dimensional (3D) reconstruction, to enhance the visualization and detection of abnormalities in breast tissue. The system builds upon the methodology proposed in [48], advancing it by incorporating a low-cost design capable of generating 3D reconstructions to provide detailed spatial information about abnormalities. This innovation addresses a critical need in medical diagnostics, particularly for conditions such as breast cancer, where the precise localization of anomalies significantly impacts patient outcomes.

The importance of accurately determining the location of abnormalities is multifaceted:

Guided biopsy: Biopsy remains the gold standard for diagnosis [49]. Identifying the precise location of anomalies ensures targeted tissue sampling for analysis.

Personalized treatment: Precise localization facilitates specific treatment plans, including the selection of appropriate therapeutic techniques and surgical interventions, minimizing damage to healthy tissue [50].

Monitoring and follow-up: Accurate tumor tracking during treatment enables the evaluation of therapeutic efficacy and timely adjustments [51].

Effective surgery: Reliable localization allows surgeons to achieve optimal tumor removal with appropriate margins, reducing recurrence risks [52].

Although ultrasound-guided core needle biopsy (US-CNB) is a widely used technique for guided biopsies due to its ability to provide high-quality tissue samples [52,53], it has limitations. US-CNB may lack the detail needed to distinguish subtle tumor characteristics [54,55], and its accuracy often depends on the operator’s expertise [56,57,58]. The proposed optomechatronic system addresses these challenges by automating image capture and processing, reducing operator dependency and variability in detection. Additionally, the integration of 3D reconstruction offers superior resolution through multi-angle imaging, enabling a more comprehensive assessment of abnormalities.

A notable limitation of ultrasound is its capacity to evaluate the tumor extent and its relationship with surrounding structures, which can affect biopsy planning [59,60,61]. In contrast, the optical 3D visualization system proposed here allows for a detailed assessment of both the tumor and adjacent tissues, facilitating better planning for biopsy and surgical procedures. This capability enhances traditional ultrasound applications by providing complementary high-resolution volumetric data.

Furthermore, optical techniques, such as DOM, enable continuous monitoring without the risks associated with ionizing radiation, making them suitable for repeated use [62,63]. Unlike mammography, which can struggle to detect tumors in dense breast tissue due to overlapping structures [64,65,66], the proposed system enhances detection in such cases. By modulating irradiation intensity and leveraging volumetric imaging, it improves the visualization of obscured lesions, thus addressing a critical gap in current diagnostic methods.

### 4.1. Strengths and Current Limitations

While the proposed system demonstrates promise in improving the resolution and detail of breast tissue imaging, several limitations need to be addressed:

Quantitative analysis deficiency: The study lacks extensive quantitative comparisons with established techniques such as mammography, ultrasound, or MRI. Metrics like sensitivity, specificity, and resolution improvement remain underexplored. A more rigorous evaluation is required to substantiate the proposed system’s efficacy.

3D surface imaging constraints: Although 3D reconstruction provides valuable spatial data, its benefit in DOM imaging is somewhat latent. The system currently captures detailed surface information but could be enhanced to offer deeper insights into internal tissue layers, bridging the gap between DOM and MRI in volumetric imaging.

Signal-to-noise ratio (SNR): Optical techniques like DOM are inherently susceptible to scattering and absorption in dense tissue. This limits their ability to distinguish subtle anomalies compared with MRI or contrast-enhanced ultrasound.

Depth penetration: Unlike ultrasound or MRI, which can visualize deeper tissue structures, optical methods are currently limited in their penetration depth. Enhancing the system’s capability to probe deeper layers will be a critical area for future research. Nevertheless, optical techniques are rapidly evolving and offer significant potential for complementing traditional diagnostic methods by providing unique contrast mechanisms, non-invasive measurements, and real-time imaging capabilities, which could lead to improved accuracy and earlier detection in clinical settings.

To avoid the limitation of depth penetration, a viable approach is to investigate the application of advanced optical techniques, such as phase-sensitive optical coherence tomography (POCT), or the implementation of alternative wavelength ranges, such as infrared, which may facilitate deeper tissue penetration without compromising spatial resolution. Moreover, the integration of hybrid methodologies combining diffuse optical microscopy (DOM) with ultrasound or optical magnetic resonance technologies holds promise for enhancing the penetration depth and providing a more comprehensive understanding of internal tissue structures. In regions of increased tissue density, an effective solution could involve the implementation of sophisticated signal processing algorithms to optimize the signal-to-noise ratio (SNR), employing techniques such as adaptive filtering or wavelength multiplexing. Finally, conducting a detailed evaluation of the system’s performance across various tissue conditions, utilizing phantoms with diverse properties, is crucial to ascertain the system’s maximum performance and its overall efficacy.

Regarding the limitation of increasing the exposure level in images, this can lead to saturation where pixels reach their maximum value, resulting in the loss of critical de-tails, especially in areas of interest such as abnormal structures. This saturation negatively impacts image quality by turning overexposed areas white, thus hindering the precise identification of key features. To address this, it is essential to thoroughly analyze the balance between lighting intensity and exposure time.

The analysis should consider how lighting affects the amount of light reaching the camera sensor and how exposure time influences light accumulation. Excessive lighting or exposure time can saturate bright areas, while insufficient lighting may cause underexposure, obscuring the anomalies. Therefore, fine adjustments to both parameters are crucial to ensure optimal contrast, preserving detailed information about abnormal structures. Experimental tests should be conducted to assess various exposure and lighting combinations to identify the best configuration for each scenario.

The relationship between exposure intensity and image saturation is a critical factor in image acquisition and processing. Exposure intensity, defined as the amount of light incident on a sensor or film, must be precisely regulated to mitigate adverse effects. Insufficient exposure results in underexposed images with a loss of information in low-luminance regions, whereas excessive exposure leads to saturation, thereby limiting the sensor’s ability to resolve details in high-intensity areas. Saturation occurs when the sensor reaches its light detection threshold, preventing the accurate discrimination of signal variations [67]. In medical imaging applications, such as fluorescence imaging and optical tomography like DOM, precise exposure calibration is essential to prevent saturation in anatomically relevant structures, thus enhancing anomaly detection and improving diagnostic reliability [68].

Image quality can be enhanced through the implementation of optical filters and polarizers, which contribute to contrast improvement. Additionally, advanced processing techniques, such as High Dynamic Range (HDR) imaging, facilitate the fusion of multiple exposures to enhance detail visibility. Furthermore, precise control of light distribution and regulation minimizes the occurrence of over-saturated regions, promoting greater uniformity in data acquisition. Proper management of sensor sensitivity and exposure time is also crucial, as elevated sensor sensitivity values increase image noise, while extended exposure durations enable the capture of additional information without amplifying the intensity of the light source [69].

### 4.2. Comparison with Other Techniques

When compared with widely used imaging methods, the optomechatronic system has unique strengths and distinct challenges:

Mammography: Mammography provides excellent resolution for calcifications and small lesions but struggles with dense breast tissue, where overlapping structures obscure abnormalities [64,65,66]. The proposed system mitigates this by offering volumetric imaging, enhancing lesion detection. However, mammography’s established metrics (e.g., 85–90% sensitivity in non-dense breasts) surpass the currently unvalidated performance of the proposed system.

Ultrasound: Ultrasound-guided biopsy (e.g., US-CNB) excels in distinguishing solid and cystic structures and provides real-time imaging for interventions [52,53]. However, its resolution and operator dependency remain limitations [56,57]. The optomechatronic system offers automation and potentially higher spatial resolution, reducing human error.

MRI: Magnetic resonance imaging offers unparalleled soft tissue contrast and depth penetration, with sensitivity often exceeding 90% for detecting malignancies in high-risk populations [70]. Nonetheless, its high cost, limited accessibility, and reliance on contrast agents pose significant barriers. In contrast, the proposed system aims to provide a low-cost alternative without ionizing radiation or invasive agents, albeit at the expense of depth resolution.

Optical systems: Like diffuse optical mammography, techniques such as fluorescence spectroscopy with indocyanine green (ICG), diffuse optical spectroscopy (DOSI), and multispectral imaging have shown promising results. ICG, for example, excels in tumor detection through fluorescence, offering high sensitivity, though its use may be limited by scattering in turbid tissues, which affects image accuracy in advanced stages [71]. DOSI has proven effective in predicting chemotherapy responses in breast cancer, but its application is constrained by the need for patient-specific calibration [72]. Fluorescence within the optical window (600–1300 nm) improves image contrast for visualizing small tumors, though tissue inhomogeneities can reduce its efficiency and reliability [73]. Multispectral imaging with LEDs enhances the molecular analysis of breast tissue, proving useful for early detection, yet costs and technical complexity remain barriers to its large-scale use [74]. Fluorescence optical spectroscopy also shows promise in distinguishing normal from tumor tissues, though real-time imaging challenges persist in clinical settings [75].

The integration of these techniques with the developed system could enhance the optical-based breast cancer diagnostic methodology, improving both accuracy and efficiency. Moreover, this combination could reduce both the costs and challenges associated with the widespread implementation of these technologies.

Finally, considering the results show in Table 3, ultrasound has the highest sensitivity (83%), indicating that it is more effective at detecting cases with anomalies (lower false negative rate). DOM with 3D reconstruction (72%) has intermediate sensitivity, which is better than that of mammography (67.8%) but lower than that of ultrasound.

Mammography shows the highest specificity (75%), meaning that it has a lower false positive rate compared with the other techniques. DOM with 3D reconstruction exhibits moderate specificity (68%), which is higher that of than ultrasound (34%) but not as high as that of mammography.

In addition to these results, a conventional mammography of the phantom was performed to visually compare the findings with the images obtained using the developed system. Figure 10 shows the mammogram of the phantom: section A presents a frontal view, considering the phantom from a top–down perspective, as performed by the developed system, while section B shows the mammogram in the conventional position used in clinical studies.

As observed, in the traditional perspective, abnormalities are not distinguishable compared with the top–down view. However, the top–down view presents false positives and a spatial displacement of the tumors from their actual position due to the pressure exerted on the tissue during the procedure.

These results demonstrate that, compared with a conventional study, diffuse optical mammography (DOM) is a promising technique that could aid in breast cancer diagnosis and enhance traditional methods.

Additionally, an ultrasound of the phantom was conducted to compare the images obtained with this technique against the results of the proposed DOM method. Figure 11 shows an ultrasound where the letter “L”, observed at the bottom, indicates the presence of a lesion. However, false positives were observed, potentially due to the phantom material. This emphasizes the crucial role of medical expertise in determining the presence of abnormalities and deciding whether further testing or supplementary diagnostic procedures are required.

Finally, Figure 12 presents an image obtained through the developed DOM system, illustrating the distinction in how lesions are perceived using non-invasive optical techniques. This image highlights the advantages of optical methods in visualizing abnormal tissue, offering a comparison with traditional imaging modalities.

The use of DOM in combination with 3D reconstruction images as non-invasive optical technique provides a promising alternative to enhancing diagnostic accuracy while minimizing patient discomfort, the costs of the analysis, and the need for more invasive procedures.

### 4.3. Future Directions: A Quantitative Perspective

To advance the system’s potential, future research should emphasize the following [76,77]:

Benchmarking against standards: A comprehensive comparison with established modalities (e.g., sensitivity, specificity, resolution) is essential. Quantitative studies involving larger datasets and diverse patient profiles can establish performance benchmarks and validate clinical applicability.

Enhancing data acquisition: Optimizing the optical system to improve the SNR and depth penetration could expand its application scope. Incorporating advanced algorithms for diffuse optical mammography (DOM) reconstruction may yield more accurate subsurface imaging.

Hybrid approaches: Combining DOM with other modalities (e.g., ultrasound or MRI) could create a hybrid system that leverages the strengths of each technique. For example, DOM’s real-time, non-invasive imaging could complement MRI’s depth resolution for comprehensive diagnostics.

Longitudinal studies: Conducting studies to monitor lesion progression over time can provide quantitative insights into the system’s role in early detection, treatment planning, and post-treatment follow-up. Metrics such as tumor volume change, margin delineation accuracy, and recurrence rates would offer valuable evidence.

Cost-effectiveness analysis: From an implementation perspective, the system’s affordability and portability make it well-suited for resource-limited settings. However, a quantitative cost–benefit analysis in comparison with mammography and ultrasound is necessary to highlight its economic viability.

The proposed optomechatronic system represents a promising alternative to traditional imaging techniques, particularly in scenarios where cost, portability, and non-invasiveness are critical. Addressing the limitations through rigorous quantitative analyses and leveraging hybrid approaches will solidify its role as a complementary tool in breast cancer diagnostics. By improving its imaging capabilities and validating its performance, the system has the potential to become an asset in advancing early detection and personalized treatment strategies.

## 5. Conclusions

In the detection of breast cancer tumors, selecting an appropriate imaging technique is essential for ensuring precise and effective evaluation. Conventional mammography has historically been the primary method for early breast cancer diagnosis, providing substantial resolution and a well-established approach for detection. However, it has notable limitations, particularly in visualizing dense breast tissue and in the detailed assessment of tumor extent. Conversely, optomechatronic systems for three-dimensional (3D) reconstruction, such as the one presented in this study, represent a promising advancement in breast imaging technology. This system offers the ability to generate detailed 3D images, which enhances the accuracy of detecting small tumors, reduces operator dependence, and eliminates exposure to ionizing radiation. Furthermore, the comprehensive evaluation afforded by 3D reconstruction supports improved planning for biopsies and treatments, thereby augmenting the efficacy of breast cancer management.

In conclusion, while mammography continues to be a valuable tool in breast cancer detection, optomechatronic 3D reconstruction systems provide significant advantages in terms of resolution, safety, and early detection capabilities. The integration of these innovative technologies into clinical practice has the potential to substantially enhance diagnostic precision and patient outcomes.

## Figures and Tables

**Figure 1 diagnostics-15-00584-f001:**
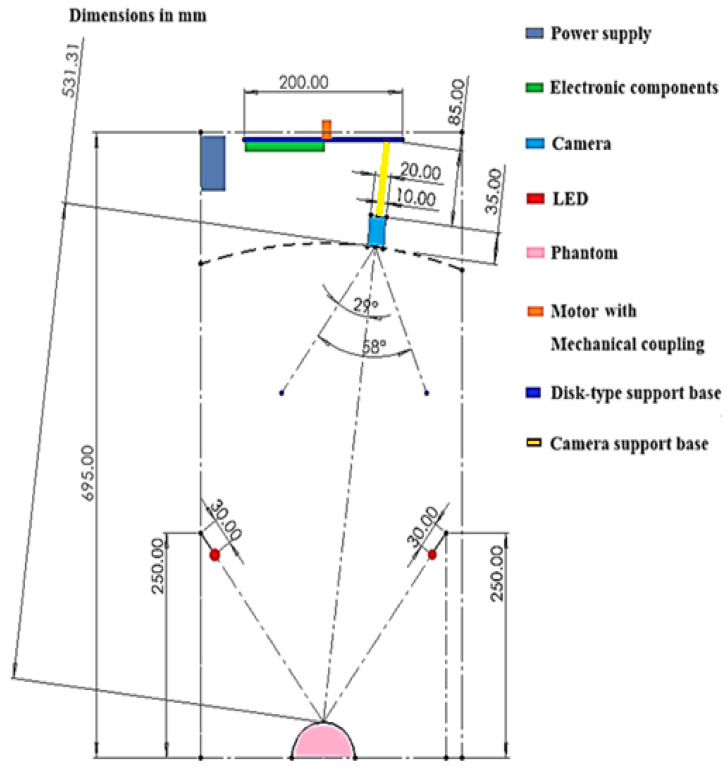
General schematic view of the system.

**Figure 2 diagnostics-15-00584-f002:**
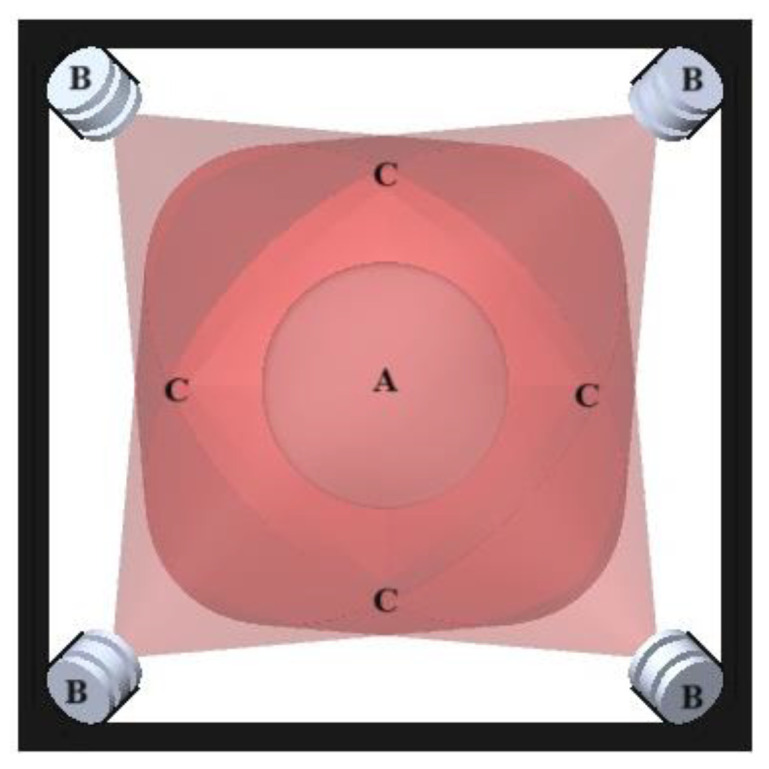
Light source arrangement. (**A**) Phantom tissue. (**B**) LED with heatsink. (**C**) Homogeneous irradiation area.

**Figure 3 diagnostics-15-00584-f003:**
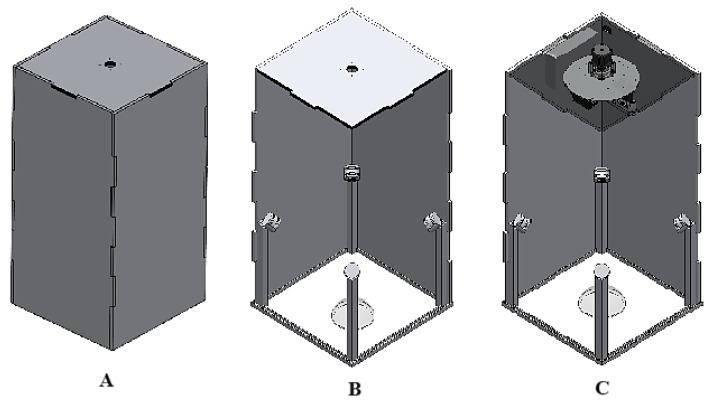
Physical structure of the system. (**A**) Booth structure. (**B**) Arrangement of the LED supports. (**C**) Disk-type support base.

**Figure 4 diagnostics-15-00584-f004:**
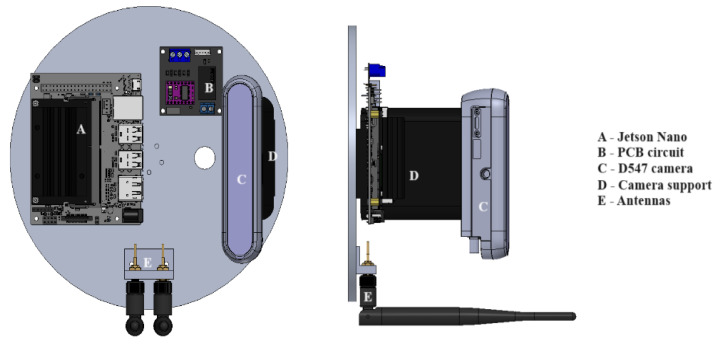
Image capture and electronic modules fixed on disk-type support base.

**Figure 5 diagnostics-15-00584-f005:**
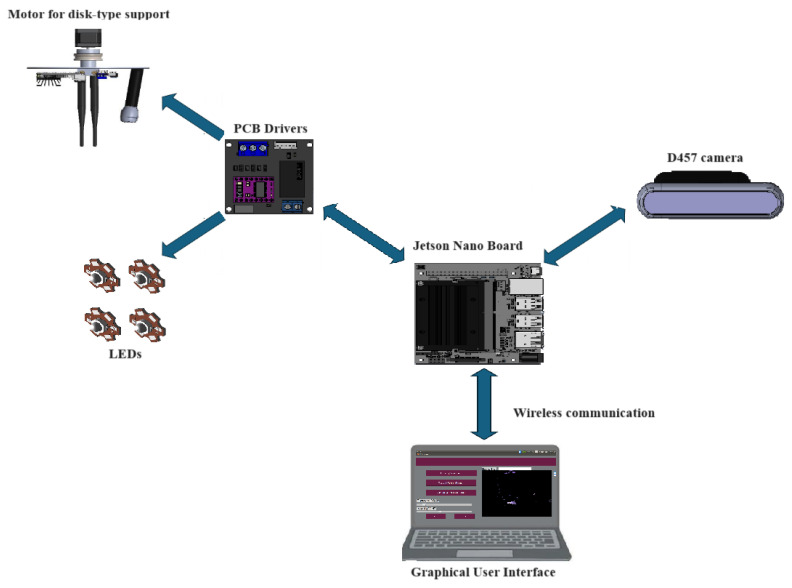
Electrical system connections.

**Figure 6 diagnostics-15-00584-f006:**
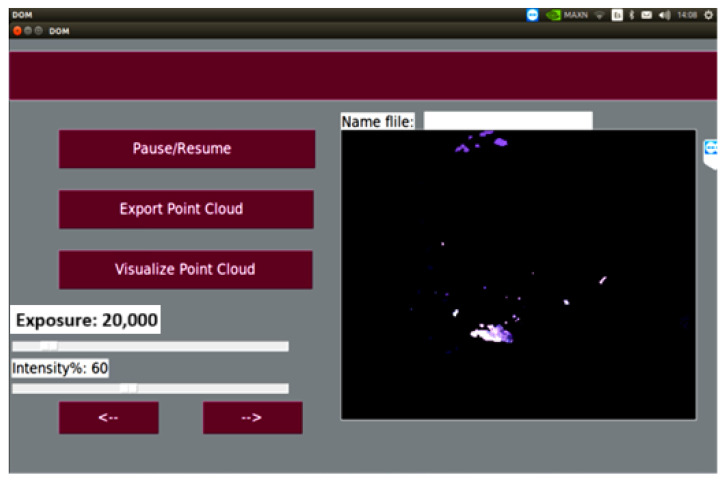
Graphical user interface.

**Figure 7 diagnostics-15-00584-f007:**
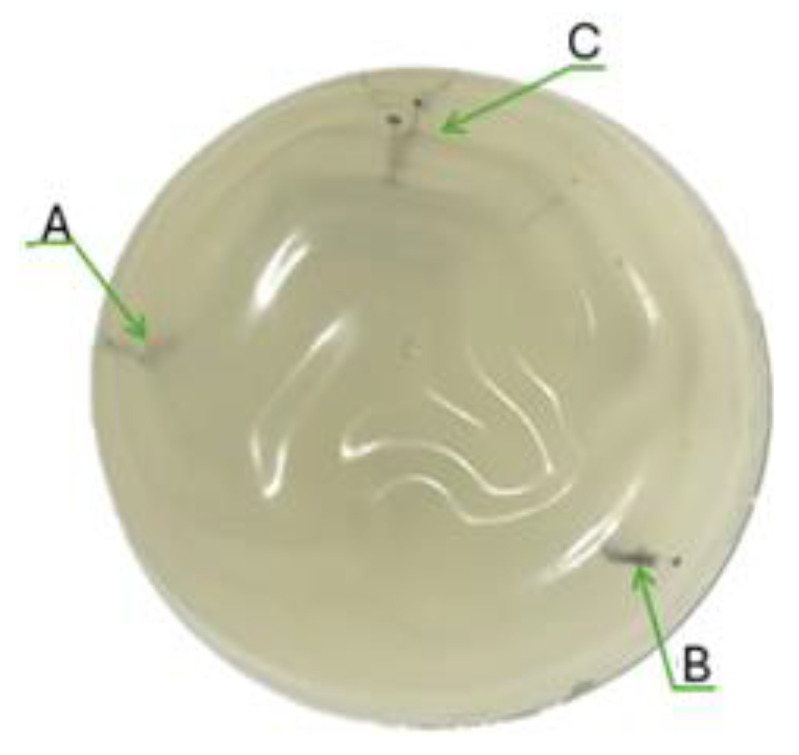
Phantom with tumor simulations of diameters (**A**) 10 mm, (**B**) 15 mm, and (**C**) 20 mm.

**Figure 8 diagnostics-15-00584-f008:**
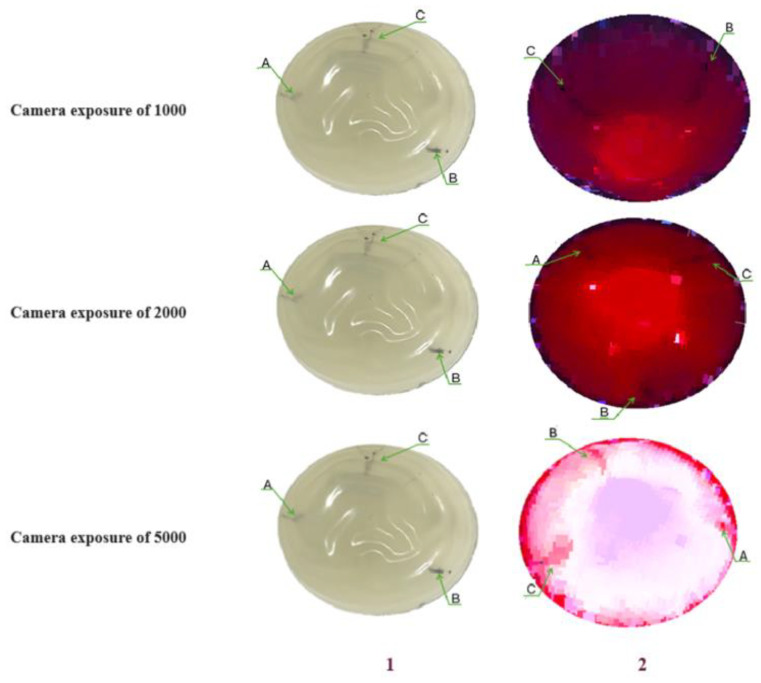
Tests using different camera exposures, an irradiation light source at 50%, and different rotations. (**1**) Phantom. (**2**) Phantom-irradiated tests. Letters A, B and C represent de tumor diameters: 10 mm, 15 mm and 20 mm respectively.

**Figure 9 diagnostics-15-00584-f009:**
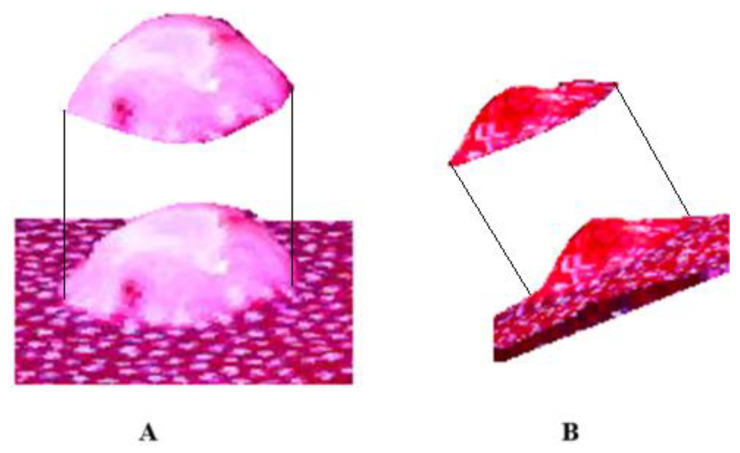
Three-dimensional reconstruction with feature extraction through 3D model visualization software. (**A**) Phantom with an exposure of 5000. (**B**) Phantom with an exposure of 1000.

**Figure 10 diagnostics-15-00584-f010:**
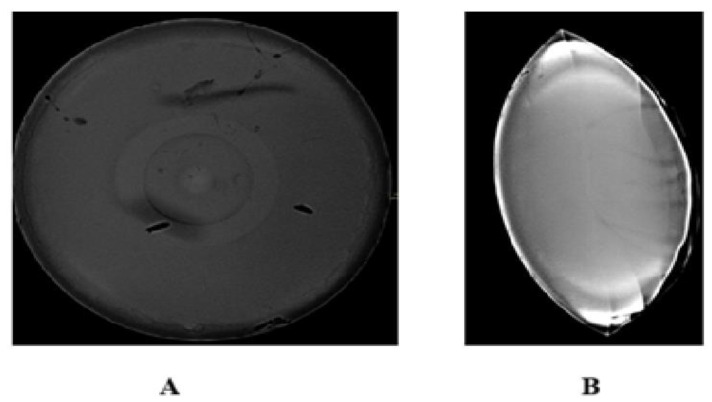
Phantom mammography. (**A**) Top view, resembling the one displayed by the developed system, and (**B**) traditional view using mammography analysis.

**Figure 11 diagnostics-15-00584-f011:**
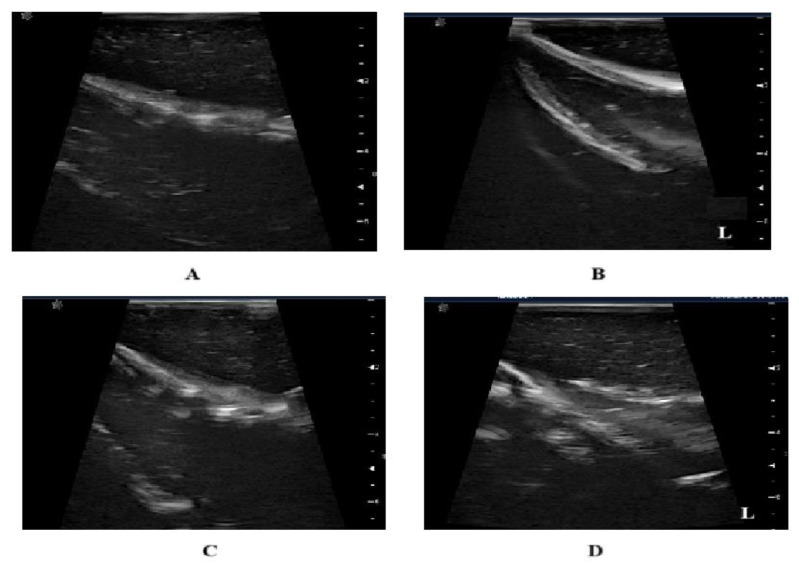
Phantom ultrasound. (**A**) True negative. (**B**) False positive. (**C**) False negative. (**D**) True positive.

**Figure 12 diagnostics-15-00584-f012:**
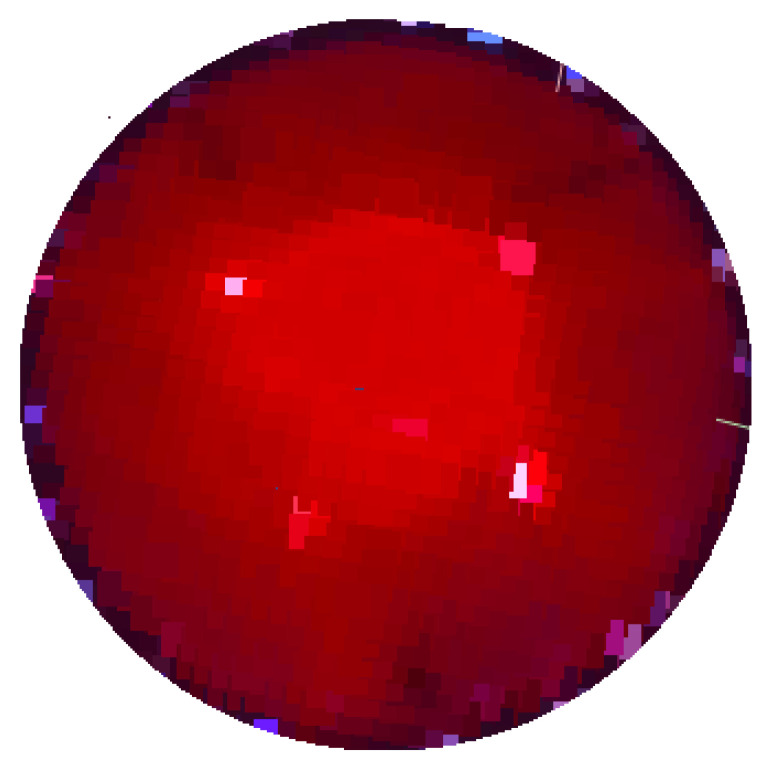
Phantom image obtained using the developed DOM system.

**Table 1 diagnostics-15-00584-t001:** Optical techniques for medical diagnosis. Data compiled from [14,15,16,17,18].

Optical Technique	Principle of Operation	Advantages over Conventional Techniques
Optical Spectroscopy	Analyzes light absorption and scattering to identify biomarkers.	Non-invasive, enables biochemical analysis without biopsies.
Diffuse Optical Tomography (DOT)	Uses near-infrared light to reconstruct tissue optical properties.	Provides depth-resolved imaging, assesses cerebral oxygenation.
Optical Coherence Tomography (OCT)	Employs low-coherence interferometry to generate cross-sectional images.	High-resolution, non-invasive, widely used in ophthalmology.
Raman Spectroscopy	Measures inelastic scattering to detect molecular composition.	High molecular specificity, useful in oncological diagnosis.
Fluorescence Imaging	Uses fluorophores to highlight specific molecules in tissues.	High sensitivity enables real-time imaging of biological processes.
Photoacoustic Imaging (PAT)	Combines laser-induced ultrasound with optical absorption contrast.	Functional imaging of vascularization and tissue oxygenation.

**Table 2 diagnostics-15-00584-t002:** Optical property comparisons between real tissue and phantoms. Data obtained from [39,40].

Material	Refractive Index at 635 nm
Real breast tissue	Fat: ~1.457 Water: ~1.332 Mixture: ~1.375–1.405
Breast implant ^1^	Silicone: ~1.406 Saline solution: ~1.332

^1^ These values are approximations and may vary depending on the material composition and measurement conditions.

**Table 3 diagnostics-15-00584-t003:** Sensitivity and specificity of the system, as well as its comparison with conventional imaging techniques such as mammography and ultrasound.

Technique	Sensitivity	Specificity
Mammography ^1^	67.8%	75%
Ultrasound ^1^	83%	34%
DOM with 3D image reconstruction	72%	68%

^1^ Data obtained from [43].

## Data Availability

The original contributions presented in the study are included in the article; further inquiries can be directed to the corresponding authors.

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
