# Peer review of "A Low-Cost Optomechatronic Diffuse Optical Mammography System for 3D Image Reconstruction: Proof of Concept"

_diagnostics, 2025, doi:10.3390/diagnostics15050584_

Round 1

Reviewer 1 Report

Comments and Suggestions for Authors

The paper presents the development research and initial test results of an opto-mechanical system for three-dimensional (3D) image reconstruction to identify abnormalities in breast tissue and support breast cancer diagnosis. The paper shows the authors' efforts in exploiting optical techniques in diagnosis when this technique often produces blurred images due to strong scattering in biological tissue. Similar published studies focus on hand-held breast light device or vascular imaging. The novelty of this study lies in the combination of diffuse optical mammography (DOM) and 3D image reconstruction, allowing the creation of detailed spatial images, supporting the early detection of abnormalities in breast tissue. The system is designed with the goal of reducing dependence on traditional diagnostic methods based on X-rays or ultrasound, which brings great value in the development of diagnostic applications. The paper needs the following important revisions:

First, the current experiment is limited to only one phantom. The author does not have quantitative analysis data to demonstrate the effectiveness and reliability of the system.

Second, the study does not provide quantitative analysis such as sensitivity, specificity, or other important parameters to compare the effectiveness of the system with current techniques. This reduces the practical applicability and competitiveness of the technology compared to standardized methods such as MRI or ultrasound.

Third, the limitations of light penetration depth in dense tissues and signal-to-noise ratio need to be addressed. The optical efficiency of the LED when interacting with the phantom and the ratio of light energy absorbed and scattered through thick tissues need to be provided. The study does not provide specific formulas or models for calculation. Back-diffusion imaging techniques are often limited by the depth of light penetration. The authors should provide information on the maximum penetration depth achieved by the system.

Fourth, regarding the reconstruction image: the spatial resolution, location, size, and depth relative to the surface of the abnormalities in the reconstructed 3D image should be discussed. The experimental results show that the system can detect two larger tumors (15 mm, 20 mm) at a low exposure level (exposure = 1000) and detect all three tumors at a higher exposure level (exposure = 2000). However, increasing the exposure level may lead to saturation, which affects the detailed information about the abnormal structure in the image. The choice between illumination intensity and exposure should be discussed further.

Author Response

Thank you very much for taking the time to review this manuscript. Please find detailed responses below and the corresponding revisions/corrections highlighted/in track changes in the re-submitted files.

Reviewer #1

The paper presents the development research and initial test results of an opto-mechanical system for three-dimensional (3D) image reconstruction to identify abnormalities in breast tissue and support breast cancer diagnosis. The paper shows the authors' efforts in exploiting optical techniques in diagnosis when this technique often produces blurred images due to strong scattering in biological tissue. Similar published studies focus on hand-held breast light device or vascular imaging. The novelty of this study lies in the combination of diffuse optical mammography (DOM) and 3D image reconstruction, allowing the creation of detailed spatial images, supporting the early detection of abnormalities in breast tissue. The system is designed with the goal of reducing dependence on traditional diagnostic methods based on X-rays or ultrasound, which brings great value in the development of diagnostic applications. The paper needs the following important revisions:

  1. First: The current experiment is limited to only one phantom. The author does not have quantitative analysis data to demonstrate the effectiveness and reliability of the system.

We understand the reviewer’s concern. We have included the information about using only one phantom for this study, but at the same time we mentioned the number of samples and described the process of the different tests that we had done. Please see Results section of the manuscript (page 9, lines 286-299).

  1. Second, the study does not provide quantitative analysis such as sensitivity, specificity, or other important parameters to compare the effectiveness of the system with current techniques. This reduces the practical applicability and competitiveness of the technology compared to standardized methods such as MRI or ultrasound.

We considered the Reviewer's comment, and we have included the specificity and sensitivity of our system and compared mammography and ultrasounds techniques values. Also, a discussion about the results was included. Please see the Results section of the manuscript (pages 10 and 11, lines 323-351).

  1. Third, the limitations of light penetration depth in dense tissues and signal-to-noise ratio need to be addressed. The optical efficiency of the LED when interacting with the phantom and the ratio of light energy absorbed and scattered through thick tissues need to be provided. The study does not provide specific formulas or models for calculation. Back-diffusion imaging techniques are often limited by the depth of light penetration. The authors should provide information on the maximum penetration depth achieved by the system.

Following the Reviewer’s suggestion, data and mathematical models were added to clarify the information and the analysis for the optical properties of the phantom. Also, the explanation of the mathematical models used.  Please see pages 10 and 11 (lines 286 – 321). Specifically: for optical efficiency of the LED when interacting with the phantom line 296-297. For the ratio of light energy absorbed and scattered through thick tissues lines 307-318. Models for calculation lines 299-321. The depth of light penetration in lines 319-321.

About the limitations of light penetration depth in dense tissues and signal-to-noise ratio, a discussion was added to explain the limitations and the possible strategies to avoid it. Please see Section 4.1 page 14 lines 447-449 and lines 457-482.

  1. Fourth, regarding the reconstruction image: the spatial resolution, location, size, and depth relative to the surface of the abnormalities in the reconstructed 3D image should be discussed. The experimental results show that the system can detect two larger tumors (15 mm, 20 mm) at a low exposure level (exposure = 1000) and detect all three tumors at a higher exposure level (exposure = 2000). However, increasing the exposure level may lead to saturation, which affects detailed information about the abnormal structure in the image. The choice between illumination intensity and exposure should be discussed further.

We really appreciate the Reviewer’s suggestion, in this sense, an explication was added to discuss the spatial resolution, location, size, and depth relative to the surface of the abnormalities in the reconstructed 3D image. Please see Results sections page 12 lines 377-388.

Reviewer 2 Report

Comments and Suggestions for Authors

       This manuscript has to do with the introduction and description of a novel diffuse optical mammography system. Diffuse optical mammography arises as an alternative to classical mammography and 3D tomosynthesis. This method offers a comfortable low-cost breast examination, avoids any breast x-ray irradiation and is capable of 3D representation of the breast. However, this imaging technique is still under experimental evaluation because it presents an inadequate image resolution .

       This paper presents a novel diffuse optical tomography configuration, describes thorough the system geometry, while introducing a simple image processing methodology.

       The described research is novel and relevant to the field of Biomedical Engineering. It is also a scientific work that arises an adequate interesting to the section of biomedical imaging sensors. The theme of diffuse optical tomography system is not so popular but stands for a significant research topic. The specific research can be extended to combined diffuse optical tomography imaging and ultrasound or magnetic resonance tomography (MRI) applications, which represent low risk examinations.

       The abstract describes in brief diffuse optical tomography system’s configuration, explains the methodology the authors used and the novelty they introduced in a simple and clearance manner. Results are presented as preliminary imaging tests with the aforementioned system.

       In the Introduction section the authors describe shortly breast cancer medical management worldwide and specifically for Mexico. They also give a rough description of mammography drawbacks. They conclude in a small paragraph on the advantageous potential inclusion of diffuse optical tomography in clinical practice. I suggest the authors should include a more analytical description of optical techniques (Principle of operation and documentation of their advantages over existing clinical applications)

       In the section Materials and Methods, the authors describe in a simple and a  good explanatory manner system’s geometry and configuration, its electromechanical parts, the electronic cards for image processing that they are attached to the camera, phantoms that they used and the software for system’s wireless handling and image processing and further  analysis. Diffuse optical tomography comprises an important biomedical research area. However, is not so widespread research field.  Perhaps the inclusion of a brief paragraph, in Materials and Methods section, on explaining and describing light interactions with breast tissue constituents, is preferable.

       In section “4.1 Strengths and Current Limitations”, I believe the authors should try to present an applicable solution on extending penetration depth and discuss on possible system’s performance optimization.

       Although diffuse optical mammography systems lack imaging resolution and possible findings quantification, the authors could present a fused optical image with a mammogram or ultrasound image of their phantom, to strengthening the purpose of their research, as far as their choice on investigating optical imaging techniques concerns.

       In section “4.2 Comparison with Other Techniques”, the authors could comment on other existing experimental optical systems.

       The manuscript is written in a very good English. It deals with a modern scientific research problem. The methodology the authors used is written in a simple, scientific and very explanatory manner.

       References are appropriate and updated. Figures can easily be read, while tables present results in clusters and cannot be misunderstood. Only on fig 1, power supply and mechanical coupling are not apparent on the presented general schematic view of the system.

       On line 248, I suggest to include a reference on the use of Indian Ink.

Author Response

Responses to the Editor and Reviewers’ Comments – Manuscript diagnostics-3465461

Thank you very much for taking the time to review this manuscript. Please find detailed responses below and the corresponding revisions/corrections highlighted/in track changes in the re-submitted files.

Reviewer #2

This manuscript has to do with the introduction and description of a novel diffuse optical mammography system. Diffuse optical mammography arises as an alternative to classical mammography and 3D tomosynthesis. This method offers a comfortable low-cost breast examination, avoids any breast x-ray irradiation and is capable of 3D representation of the breast. However, this imaging technique is still under experimental evaluation because it presents an inadequate image resolution.

       This paper presents a novel diffuse optical tomography configuration, describing geometry thorough the system, while introducing a simple image processing methodology.

       The research described is novel and relevant to the field of Biomedical Engineering. It is also scientific work that arises an adequate interest in the section of biomedical imaging sensors. The theme of diffuse optical tomography system is not so popular but stands for a significant research topic. The specific research can be extended to combined diffuse optical tomography imaging and ultrasound or magnetic resonance tomography (MRI) applications, which represent low risk examinations.

       The abstract describes in brief a diffuse optical tomography system’s configuration, explains the methodology the authors used and the novelty they introduced in a simple and clear manner. Results are presented as preliminary imaging tests with the system.  

  1. In the Introduction section the authors describe shortly breast cancer medical management worldwide and specifically for Mexico. They also give a rough description of mammography drawbacks. They conclude in a small paragraph on the advantageous potential inclusion of diffuse optical tomography in clinical practice. I suggest the authors should include a more analytical description of optical techniques (Principle of operation and documentation of their advantages over existing clinical applications).

As Reviewer pointed out, analytical description of optical techniques (principle of operation and documentation of their advantages over existing clinical applications) was added, please see page 2 in lines 58-72.

  1. In the section Materials and Methods, the authors describe in a simple and a good explanatory manner the system’s geometry and configuration, its electromechanical parts, the electronic cards for image processing that they are attached to the camera, phantoms that they used and the software for system’s wireless handling and image processing and further analysis. Diffuse optical tomography comprises an important biomedical research area. However, it is not so widespread research field.  Perhaps the inclusion of a brief paragraph, in the Materials and Methods section, on explaining and describing light interactions with breast tissue constituents, is preferable

Following the Reviewer’s suggestion, we apologize for a pour explanation about this section, a brief explanation and description of the light interaction with breast tissue was added. Please see Material and Methodes section on page 4 at lines 109-119.    

  1. In section “4.1 Strengths and Current Limitations”, I believe the authors should try to present an applicable solution on extending penetration depth and discuss on possible system’s performance optimization.

Following the Reviewer’s suggestion, now we add the discussion about to present an applicable solution on extending penetration depth. We have included clarification concerning these points on page 14 lines 457-482.   

  1. Although diffuse optical mammography systems lack imaging resolution and possible findings quantification, the authors could present a fused optical image with a mammogram or ultrasound image of their phantom, to strengthening the purpose of their research, as far as their choice on investigating optical imaging techniques concerns.

We sincerely appreciate the reviewer’s suggestion. To attend the comment, images from mammography and ultrasound of the phantom were added to the manuscript on section 4.2 to compare the images with applied traditional analysis and the images with the DOM developed system. Please see section 4.2, pages 15, 16 and 17 lines 528-564.

  1. In section “4.2 Comparison with Other Techniques”, the authors could comment on other existing experimental optical systems.

Following the Reviewer’s suggestion, other existing experimental optical systems were mentioned and compared with the presented system. Please see page 15, lines 503-527.

  1. The manuscript is written in a very good English. It deals with a modern scientific research problem. The methodology the authors used is written in a simple, scientific and very explanatory manner.

We appreciate the comment of the reviewer, and the time to analyze our manuscript language.

  1. References are appropriate and updated. Figures can easily be read, while tables present results in clusters and cannot be misunderstood. Only fig 1, power supply and mechanical coupling are not apparent on the presented general schematic view of the system.

We really apologize because of the mistake, Figure 1 was corrected, please see the figure on page 3 line 103.

  1. On line 248, I suggest including a reference on the use of Indian Ink.

As Reviewer pointed out, a reference about the use of Indian Ink was added. Please see the Results section on page 8 in lines 268-269.

Round 2

Reviewer 1 Report

Comments and Suggestions for Authors

The revised manuscript shows clear improvements, particularly in including sensitivity (72%) and specificity (68%) values and a more detailed discussion of optical properties and penetration depth. The explanation of spatial resolution, anomaly size, and depth in the reconstructed 3D images is clearly articulated, and the analysis of exposure intensity and its impact on image saturation adds clarity to the system's performance. Additionally, an expanded discussion of light-tissue interactions and signal-to-noise ratio constraints reinforces the scientific foundation of the manuscript.
Some points could be improved, such as discussing the trade-off between exposure intensity and saturation, which would benefit from more concise wording to enhance readability. Additionally, while comparing images reproduced at different exposure levels is insightful, emphasizing the optimal balance between illumination intensity and image clarity would have provided more practical relevance. With these minor adjustments, the manuscript would be well-positioned for publication.

Author Response

Thank you very much for taking the time to review this manuscript. Please find detailed responses below and the corresponding revisions/corrections highlighted/in track changes in the re-submitted files.

  1. Some points could be improved, such as discussing the trade-off between exposure intensity and saturation, which would benefit from more concise wording to enhance readability. Additionally, while comparing images reproduced at different exposure levels is insightful, emphasizing the optimal balance between illumination intensity and image clarity would have provided more practical relevance. With these minor adjustments, the manuscript would be well-positioned for publication.

We understand the reviewer’s concern. We have included the information about the trade-off between exposure intensity and saturation emphasizing the optimal balance between illumination intensity and image clarity. Please see Discussion section 4.1 (page 15, lines 496-515).

Reviewer 2 Report

Comments and Suggestions for Authors

The authors have answered to all my suggestions. I believe the manuscript is enriched with additionally usefull to the reader information. 

Some lines throughout the manucripth have been altered (I did not comment on them, on my first review): 286-351, 377-388, 447-449, 503-527, 774-786. However the inclusion of these lines are completely justified. 

Finally, according to manuscript's data, the penetration depth (line 320) seems to be a little overestimated. The reference [24] concludes on a value of 0.09 for the absorbing coefficient of silicone polymer samples that are enriched with  aluminum powder. Perhaps the authors could substantiate better the value they found with an additional bibliographic reference. 

References are not numbered correctly on section: References

Author Response

Thank you very much for taking the time to review this manuscript. Please find detailed responses below and the corresponding revisions/corrections highlighted/in track changes in the re-submitted files.

Reviewer #2

  1. The authors have answered all my suggestions. I believe the manuscript is enriched with additionally useful to the reader information.

We appreciate the comment of the reviewer, and the time to review our manuscript.

  1. Some lines throughout the manuscript have been altered (I did not comment on them, on my first review): 286-351, 377-388, 447-449, 503-527, 774-786. However, the inclusion of these lines is completely justified.

We appreciate the comments of the reviewer; information was added in order to improve the manuscript considering also the others reviewer’s comments.

  1. Finally, according to manuscript's data, the penetration depth (line 320) seems to be a little overestimated. The reference [24] concludes with a value of 0.09 for the absorbing coefficient of silicone polymer samples that are enriched with aluminum powder. Perhaps the authors could substantiate better the value they found with an additional bibliographic reference.

Following the Reviewer’s suggestion, now we add an additional bibliographic reference about the composition of silicone phantoms for breast tissue and its relationship with Al powder. Please see this on page 10 lines 319-325. About the penetration depth also we have included now a brief explanation of the value. Please see page 9 lines 330-335.  

  1. References are not numbered correctly on section: References

We apologize because of the mistake. References were checked and corrected.